# Gastric Enteric Glial Cells: A New Contributor to the Synucleinopathies in the MPTP-Induced Parkinsonism Mouse

**DOI:** 10.3390/molecules27217414

**Published:** 2022-11-01

**Authors:** Yang Heng, Yan-Yan Li, Lu Wen, Jia-Qing Yan, Nai-Hong Chen, Yu-He Yuan

**Affiliations:** 1State Key Laboratory of Bioactive Substances and Functions of Natural Medicines, Department of Pharmacology, Institute of Materia Medica & Neuroscience Center, Chinese Academy of Medical Sciences and Peking Union Medical College, Beijing 100050, China; 2Department of Pharmacy, National Cancer Center/National, Clinical Research Center for Cancer/Cancer Hospital, Chinese Academy of Medical Sciences and Peking Union, Medical College, Beijing 100021, China

**Keywords:** Parkinson’s disease, α-synuclein, enteric glial cells, gut-brain axis, 1-methyl-4-phenyl-1,2,3,6-tetrahydropyridine

## Abstract

Accumulating evidence has shown that Parkinson’s disease (PD) is a systemic disease other than a mere central nervous system (CNS) disorder. One of the most important peripheral symptoms is gastrointestinal dysfunction. The enteric nervous system (ENS) is regarded as an essential gateway to the environment. The discovery of the prion-like behavior of α-synuclein makes it possible for the neurodegenerative process to start in the ENS and spread via the gut-brain axis to the CNS. We first confirmed that synucleinopathies existed in the stomachs of chronic 1-methyl-4-phenyl-1,2,3,6-tetrahydropyridine (MPTP)/probenecid (MPTP/p)-induced PD mice, as indicated by the significant increase in abnormal aggregated and nitrated α-synuclein in the TH-positive neurons and enteric glial cells (EGCs) of the gastric myenteric plexus. Next, we attempted to clarify the mechanisms in single MPTP-injected mice. The stomach naturally possesses high monoamine oxidase-B (MAO-B) activity and low superoxide dismutase (SOD) activity, making the stomach susceptible to MPTP-induced oxidative stress, as indicated by the significant increase in reactive oxygen species (ROS) in the stomach and elevated 4-hydroxynonenal (4-HNE) in the EGCs after MPTP exposure for 3 h. Additionally, stomach synucleinopathies appear before those of the nigrostriatal system, as determined by Western blotting 12 h after MPTP injection. Notably, nitrated α-synuclein was considerably increased in the EGCs after 3 h and 12 h of MPTP exposure. Taken together, our work demonstrated that the EGCs could be new contributors to synucleinopathies in the stomach. The early-initiated synucleinopathies might further influence neighboring neurons in the myenteric plexus and the CNS. Our results offer a new experimental clue for interpreting the etiology of PD.

## 1. Introduction

Parkinson’s disease (PD) is a common neurodegenerative disease characterized by the loss of dopaminergic neurons in the substantia nigra (SN) and the appearance of Lewy bodies (LBs), mainly consisting of abnormal α-synuclein in the central nervous system (CNS) [1,2]. Gastrointestinal dysfunction is a well-recognized feature of PD. Various findings suggest gastrointestinal syndromes may be caused by a similar degenerative process in the enteric nervous system (ENS) [3].

It is believed that the interaction of genetic and environmental factors leads to the progression of PD and that genetic factors can increase the susceptibility to environmental factors [4]. α-synuclein is the expression product of the gene SNCA, one of the genetic factors contributing to PD. Its abnormal expression, distribution, modification, and aggregation are tightly related to PD progression [5,6]. Recent evidence has indicated that α-synuclein is a prion-like protein that could propagate from one cell to another [7,8]. Furthermore, pathologic α-synuclein aggregates are first detected in the gastrointestinal tract and submandibular glands during the prodromal stage of PD rather than in the SN [9,10], thereby leading to the hypothesis that environmental insults (i.e., toxins or pathogens) may first affect the ENS, causing local α-synuclein aggregation. The pathology in the ENS could then spread via the gut-brain axis to the CNS [11]. Recently, several scientists have confirmed this hypothesis in their labs. Pan-Montojo and co-workers showed that chronic intra-gastric administration of low doses of rotenone, leading to undetectable neuroinflammation, could trigger neurodegenerative pathology similar to PD in mice. They first discovered that enteric neurons could release α-synuclein and that resection of the autonomic nerves could stop the progression of the synucleinopathies into the CNS [12]. Furthermore, Staffan demonstrated that intra-gastrointestinal wall (stomach and duodenum) delivery of different forms of α-synuclein (human PD brain lysate and recombinant α-synuclein) could propagate from the gut to the brain in rats via the vagal nerve in a time-dependent manner [13]. However, the cause of α-synuclein aggregation in the gastrointestinal tract remains unclear. Therefore, it is essential to investigate the synucleinopathies in the gastrointestinal tract to better understand and treat PD.

α-synuclein exists in an α-helical conformation; however, when stress occurs, it transforms into the pathological β-sheet-rich conformation, which more easily forms LBs and LNs (Lewy neuritis) [14]. Oligomeric α-synuclein is regarded as one of the most toxic species of α-synuclein and can influence many physiological processes [15]. In addition to the concentration of α-synuclein itself, post-translational modifications (PTMs), such as phosphorylation, nitration, and oxidation, also contribute to the α-synuclein aggregation [16]. Even a small change in the α-synuclein modification can substantially impact the aggregation process, converting it into the disease-related form [17,18].

1-methyl-4-phenyl-1,2,3,6-tetrahydropyridine (MPTP) is one of the most extensively used neurotoxins to produce parkinsonism in animals [19]. Three MPTP regimens are commonly used to establish PD mouse models: acute, subacute, and chronic regimens [20]. Compared with the acute and subacute regimens, the chronic MPTP/probenecid (MPTP/p) model is considered to be most appropriate for studying the pathology and mechanisms of PD [21,22]. In addition to inducing primary PD symptoms and neuroinflammation, MPTP/p administration could increase α-synuclein levels in the SN of mice and monkeys [23,24,25]. Moreover, α-synuclein inclusion bodies are only found in the chronic MPTP mouse model, but not in acute or subacute models [26]. The current studies of MPTP on the ENS are based on short-period exposure, which might not be sufficient to induce neuropathological changes [27]. Therefore, we used the chronic MPTP/p-induced PD mouse model to investigate α-synuclein abnormalities in our study. We analyzed the aggregation, nitration, and phosphorylation of α-synuclein in the stomach of the chronic MPTP/p-induced PD mouse model; determined the cytokines, neurotransmitters, and inflammation-related molecules; and confirmed the most affected cells are in the stomach by immunochemistry and immunofluorescence-staining. Through the chronic PD mouse model, we found that the enteric glial cells (EGCs) in the stomach experienced the most dramatic changes in terms of α-synuclein. Secondly, we investigated the mechanisms underlying this phenomenon using the single MPTP injection model. Results showed that the stomach was more vulnerable to MPTP invasion than SN and striatum (STR) in oxidative stress and α-synuclein abnormalities. Finally, the results indicated that EGCs could be the initial cells contributing to the synucleinopathies in the stomach.

## 2. Results

### 2.1. The Chronic MPTP/p-Treated Mouse Exhibits Motor, Gastrointestinal and Typical Pathological Alterations, but Not Olfactory Deficits

The rotarod test was performed in this study to evaluate animal motor coordination (Figure 1A). Mice treated with MPTP/p displayed a significant reduction (*p* < 0.05) in rotarod performance compared to the normal controls. In the pole test (Figure 1B), MPTP/p-injected mice needed significantly prolonged time (*p* < 0.05) to climb down from the top, indicating bradykinesia. The buried pellet test was performed to assess olfactory capacity across the groups (Figure 1C). Mice treated with MPTP/p took a little longer to discover the buried pellet. Notably, the MPTP/p-treated mice exhibited significantly higher stool frequency during 2–4 h (*p* < 0.01) and 4–6 h (*p* < 0.05), demonstrating that their gastrointestinal function was abnormal after serial exposure to MPTP (Figure 1D).

Besides motor and non-motor symptoms, mice treated with MPTP/p showed typical pathological alterations. As shown in the table (Figure 1E), the DA concentration in the STR of the MPTP/p group decreased by 93% compared with that in the control mice (*p* < 0.001). Additionally, we also observed significant decreases (*p* < 0.01) in 5-HT and dopamine metabolite DOPAC and HVA concentrations in the STR. Further, we investigated the pathological changes in the nigrostriatal system via IHC, which revealed that the number of TH-positive neurons in the SN of MPTP-injected mice decreased by 93% compared with the control groups (*p* < 0.001) (Figure 1F,G). Similarly, the density of TH-positive fibers also decreased by 55% in the STR (Figure 1H,I) (*p* < 0.001).

### 2.2. Chronic Treatment with MPTP/p Could Lead to Abnormal α-Synuclein Aggregation, Represented by Increases in Different PTMs and Aggregation Forms in the Stomach

First, we determined the protein levels of α-synuclein in the stomach with an anti-α-syn (C-20) antibody. This antibody mainly recognizes three forms of α-synuclein: the protein bands at approximately 15, 30, and 38 kDa. They were all increased significantly (*p* < 0.05) after serial MPTP/p exposure (Figure 2A,C and D). To further explore the aggregation of α-synuclein in the stomach, we analyzed α-synuclein with the 5G4 antibody (Figure 2B,E). 5G4 is a new antibody with specific reactivity against disease-related human α-synuclein with superior immunohistochemical properties, confirmed in mice in our previous study [28,29]. The 5G4 antibody recognized mainly bands at approximately 70, 55, 38, 30, and 25 kDa. After serial MPTP/p injections, the oligomeric α-synuclein forms at approximately 70 (*p* < 0.01), 55 (*p* < 0.05) and 30 kDa (*p* < 0.05) in the stomach were increased significantly.

PTMs are also disease-related and aggregation-contributed [30]. α-synuclein phosphorylated at Ser129 (p-syn [S129]), and nitrated α-synuclein (n-syn) were also detected in the stomach. All of the phosphorylated and nitrated forms of α-synuclein in the stomach were increased considerably after serial MPTP/p injections, and the nitrated forms of α-synuclein at approximately 55 kDa were increased significantly (*p* < 0.05) (Figure 2C–E).

### 2.3. Chronic MPTP/p Exposure Induced Synucleinopathies in the Gastric Myenteric Plexus

We first looked into the morphological changes induced by MPTP in the stomach through HE staining (Figure 3A). In the myenteric plexus, cells in the model group were atrophied, and the myenteric plexus appeared swollen. In the submucosa, there was a dramatic increase in inflammatory infiltrating cells, indicating a severe inflammatory state in the stomach. The mucosal parietal cells were saturated in the mucosa, and the cellular outline was evident in the normal condition. However, in the stomachs of MPTP/p-treated mice, parietal cells became atrophied and metamorphosed. Because we have confirmed that the levels of α-synuclein detected by 5G4 and anti-n-syn antibody significantly increased in the stomachs of MPTP/p-injected mice, as seen by Western blotting, we next used these two antibodies to locate which parts of the stomach had the most dramatic synucleinopathy changes. For 5G4 immunostaining (Figure 3B) in the myenteric plexus, the intensity of 5G4-positive particles was considerably elevated after serial MPTP injections. Notably, the 5G4-positive particles were similar to LBs in a further magnified view (black arrow in Figure 3B). In the mucosa, in the normal condition, specific cells naturally expressed high levels of 5G4-recognized α-synuclein. These cells are similar to enteroendocrine cells in the stomach [31]. After MPTP/p injections, the intensity of 5G4-positive particles in the mucosa was also increased. For n-syn immunostaining (Figure 3C) in the myenteric plexus, the n-syn-positive staining was gathered in a series of cells. After 10 MPTP/p injections, the nitrated α-synuclein in the myenteric plexus increased considerably, and the n-syn-positive cells were similar to LBs in a further magnified view. In the mucosa, n-syn-positive staining was also possibly concentrated in the enteroendocrine cells. Compared with the standard control, the intensities of n-syn-positive particles in the mucosa of the stomachs of the model animal were considerably increased.

### 2.4. The Synucleinopathies Mainly Localized in the TH-Positive Enteric Neurons and Enteric Glial Cells (EGCs) of the Myenteric Plexus

Immunofluorescence was performed to locate the synucleinopathies in specific types of myenteric plexus cells. First, we observed the 5G4-positive α-synuclein expression in the TH-positive enteric neurons and EGCs across groups. Much more 5G4-positive α-synuclein was co-localized with glial fibrillary acidic protein (GFAP, an EGC marker) after chronic MPTP/p exposure (Figure 4A). In the enteric neurons, aggregated α-synuclein naturally existed in some TH-positive neurons. Some TH-positive neurons did not aggregate as much as in the control group in Figure 4B. However, after chronic MPTP/p injections, 5G4-positive α-synuclein significantly increased, and all TH-positive neurons expressed high levels of aggregated α-synuclein.

Next, double-labeled immunofluorescence analysis for GFAP/n-syn and TH/n-syn showed that in the control group, a few n-syn-positive particles naturally existed in the EGCs TH-positive neurons. After chronic MPTP/p exposure, the n-syn expression was increased in both the EGCs and TH-positive neurons (Figure 4C,D).

### 2.5. Chronic MPTP/p Exposure Influenced the Functions of EGCs and TH-Positive Neurons in the Stomach

We first determined the GFAP and TH expression in the stomach across groups (Figure 5A,B). After chronic MPTP/p injections, the GFAP expression levels were dramatically decreased (*p* < 0.001), while the TH expression levels in the stomach were significantly upregulated (*p* < 0.01). We also monitored the neurotransmitters in the stomach across groups (Figure 5C). Surprisingly, we found that the DA concentrations in the stomachs of MPTP/p-treated mice were increased significantly (*p* < 0.05) compared to the control animals, which might be caused by the upregulated TH expression. Notably, the dopamine metabolites HVA and DOPAC, 5-HT, and NE did not show much variation. However, the ratios of DOPAC/DA (*p* < 0.05) and HVA/DA (*p* < 0.05) were decreased significantly.

### 2.6. The Chronic Treatment with MPTP/p Led to a Prolonged Inflammatory State and the Activation of the Nuclear Factor-κB (NF-κB) Pathway

IL-1β is a typical pro-inflammatory cytokine derived from its precursor, pro-IL-1β. Its maturation relies on the presence of caspase-1. Caspase-1 also has its precursor, pro-caspase-1, which could be activated by ROS, prion protein fibrils, and other processes [32,33]. IL-1β production relied on activation by caspase-1. Western blotting results revealed that the levels of pro-caspase-1 (approximately 45 kDa) were decreased significantly after chronic MPTP/p injections (Figure 6A,B) (*p* < 0.05), while the levels of mature caspase-1 (approximately 10 kDa) significantly were increased (*p* < 0.05). Unlike caspase-1, both the pro- (*p* < 0.05) and mature (*p* < 0.05) forms of IL-1β were significantly increased in the MPTP/p-injected mice compared to control animals (Figure 6B,C). ELISA results showed that even 15 days after the last MPTP/p exposure, we still observed a considerable increase in IL-1β production (Figure 6D) and a significant increase (*p* < 0.05) in TNF-α production compared with the control animals (Figure 6E).

The production of the pro-inflammatory cytokines TNF-α and IL-1β is tightly regulated by the NF-κB signaling pathway. Western blotting results showed that the protein levels of signaling molecules such as p-IKK-α/β (*p* < 0.01), IKK-α (*p* < 0.05), p-IkB-α (*p* < 0.05), and NF-κB (*p* < 0.05) were persistently expressed at high levels in the stomachs of MPTP/p-exposed mice, even 15 days after the last MPTP exposure (Figure 6F,G).

### 2.7. Naturally High Activities of MAO-B and Low Activities of SOD Make the Stomach Appear to Undergo Oxidative Stress before the SN, Which Is Commonly Thought to Be Highly Susceptible

After confirmation of the synucleinopathies in the stomachs of MPTP/p-induced PD mice, we next attempted to identify the underlying mechanisms. MAO-B is a crucial enzyme that catalyzes MPTP into its toxic form, MPP^+^. MPP^+^ then inhibits mitochondrial complex I, leading to excessive ROS production and oxidative stress in the cells. SOD is an essential enzyme for clearing ROS and protecting cells from oxidative stress. MAO-B and SOD activities were measured across different CNS tissues, gastrointestinal organs, blood components, and other critical peripheral organs. The gastrointestinal tract naturally possessed higher MAO-B activity than other tissues (Figure 7A). Notably, the stomach had the highest MAO-B activity among these tissues. Additionally, the extremely low MAO-B activity in the plasma and RBC inclusions may indicate that most MPTP should be converted to MPP^+^ in tissues rather than in the blood. There were no apparent variations in MAO-B activities between the control and model mice in various tissues, indicating that serial MPTP injections did not influence the MAO-B activity. Next, SOD activities across tissues were also determined (Figure 7B). Interestingly, tissues from the gastrointestinal tract had the lowest SOD levels compared to those from the CNS and peripheries.

Next, we measured ROS production levels in the SN, the STR, the colon, the ventral half of the stomach, and the kidney 3 h after a single injection of MPTP (40 mg/kg) (Figure 8A). Only 3 h after MPTP injection, ROS levels in the SN of the model group increased considerably but not significantly. MPTP induced a significant increase (*p* < 0.05) of ROS production in the stomach, indicating that the stomach is more vulnerable to MPTP invasion. Furthermore, immunofluorescence results showed that the intensity of 4-hydroxynonenal (4-HNE), a biomarker for cell damage due to oxidative stress, increased in some stomach EGCs after 3 h of exposure to MPTP (Figure 8B).

### 2.8. The Stomach Is More Susceptible to MPTP Than SN and STR, and EGCs Could Be the Initial Cells Contributing to the Synucleinopathies

Only 3 h after MPTP injection, the protein levels of α-synuclein in the stomach (approximately 38 and 30 kDa) detected with the anti-α-syn (C-20) antibody were increased considerably in the model group, especially the 30-kDa form of α-synuclein (*p* = 0.057) (Figure 9A,B). 12 h after MPTP exposure, the expression levels of both the 38- and 30-kDa forms of α-synuclein were increased significantly in the stomach (*p* < 0.01) (Figure 9A,B). The expression levels of neither the oligomeric nor the monomeric forms of α-synuclein showed significant variations at these two time points in either the SN (Figure 9C,D) or the STR (Figure 9E,F). We further determined the protein levels of oligomeric, phosphorylated, and nitrated α-synuclein 12 h after MPTP injection by Western blotting (Figure 10A–D). Only the expression levels of nitrated α-synuclein (approximately 30 kDa) in the stomach were increased significantly (*p* < 0.05) after single MPTP injection. Next, the stomach slices were double-labeled with GFAP and n-syn antibodies (Figure 10E). A single MPTP injection was sufficient to induce a considerable increase in the nitrated α-synuclein levels in the gastric EGCs after 3 h and 12 h in the MPTP-exposed mice.

## 3. Materials and Methods

### 3.1. Chemical Materials

MPTP hydrochloride (MPTP-HCl), probenecid, dopamine (DA), 5-hydroxytryptamine (5-HT), norepinephrine (NE), dihydroxyphenylacetic acid (DOPAC), homovanillic acid (HVA), and isoproterenol (IP) were purchased from Sigma-Aldrich (St. Louis, MO, USA). All other chemicals and reagents used in the experiment were of analytical grade.

### 3.2. Animals

Male C57BL/6 mice (22–25 g) aged between 8 and 9 weeks were used for all experiments (Charles River Co., Beijing, China). The mice were raised on a 12-h light/dark cycle with food and water available ad libitum and were housed in groups of five per cage with a room temperature of 22 ± 1 °C. All experiments were performed by the guidelines established by the National Institutes of Health for the care and use of laboratory animals and were approved by the Animal Care Committee of the Peking Union Medical College and the Chinese Academy of Medical Sciences. Maximum efforts were made to minimize animal suffering as much as possible.

### 3.3. MPTP Administration

The chronic PD mouse model was prepared according to our previous work (Figure 1) [29]. After one week of acclimatization, the mice were pre-trained on the rotarod and the pole apparatus. Mice with motor deficits were excluded from subsequent treatment. After pre-training, the animals were randomly assigned into two groups: the control group and the model group. Mice from the model group received 10 doses of MPTP-HCl injection (25 mg/kg b.w., dissolved in 0.9% saline, subcutaneous [s.c.]) on a 40-day schedule at intervals of 4 days between consecutive injections in combination with an adjuvant drug, probenecid (250 mg/kg b.w., dissolved in Tris-HCl, intraperitoneal [i.p.], 1 h before the MPTP-HCl injection); control mice were injected with respective doses of saline and probenecid. The animals were kept warm after injection using a heating pad and a radiator to avoid excessive deaths.

The acute model was prepared with a single injection of MPTP-HCl (40 mg/kg in saline, s.c.).

### 3.4. Behavior Testing

All behavior tests were performed between 9 a.m. and 3 p.m. under standard animal room lighting. The rotarod test was performed on D43, the pole test was performed on D44, the stool collection test was performed on D45, and the buried pellet test was performed on D46–D49 (Figure 11).

#### 3.4.1. Rotarod Test and Pole Test

Motor coordination and balance were assessed using a rotarod apparatus (IITC Life Science, Woodland Hills, CA, USA) and a self-made pole. The exact method was described in our previous work [29]. The rotarod was programmed to rotate with a linearly increasing speed from 5 rpm to 30 rpm in 300 s. Then, the animals were placed on the rolling rod and subjected to three trials with an interval of at least 30 min. The mean latency time to fall off the rotarod was recorded. The pole test apparatus consisted of a wooden pole with a height of 50 cm and a diameter of 0.5 cm wrapped in gauze to prevent slipping. The base of the pole was positioned in the home cage and covered with bedding to protect the mice from injury. A wooden ball was glued on top of the pole to help position the animals on the pole. The total time required to climb down the pole was measured. Each animal performed three successive trials at 30 min intervals. The average of the three trials was calculated for statistical analysis.

#### 3.4.2. Stool Collection Test

Based on the classic stool collection test, we modified the previous method. In detail, each animal was placed in a clear plastic cage without food and water for 6 h. Every 2 h, the mice were transferred into a new plastic cage. To rule out disturbance, during each 2 h, the room was kept quiet, and visitors were prohibited. The number of stools every 2 h was noted.

#### 3.4.3. Buried Pellet Test

The buried pellet test method was adopted from previous works [34]. At 16–18 h before the test, all chow pellets from the home cage were removed. During the testing, each mouse was placed individually in a clean holding cage (25 cm L × 15 cm W × 13 cm H) for 5 min, transferred to the testing cage (46 cm L × 23.5 cm W × 20 cm H) for 2 min acclimation, and then returned to the holding cage after a pellet was buried approximately 1 cm under the bedding in a random location in the testing cage. The buried pellets were of identical size and were not visible to each mouse. Then, each mouse was placed in the center of the test cage and given 5 min to discover the pellet. Room lighting was identical for each test, and the experiment was conducted in a quiet room. The experimenter stood at least 2 m from the cage. The latency for mice to begin eating the pellet was recorded. Each animal performed 3 trials on three consecutive days (D47–D49). The average time of the last two trials was calculated for statistical analysis.

### 3.5. Tissue Preparation

Blood was collected via the abdominal aorta from chloral hydrate (400 mg/kg, i.p.)-anesthetized mice. Blood was then transferred into EDTA-coated tubes. Plasma was separated from blood cells by centrifugation at 3500× *g* for 10 min at 4 °C. Red blood cell (RBC) pellets were handled according to the previous work [35]. Briefly, the RBC pellets were subjected to 3 consecutive washes with 1:1 (*v*/*v*) volume 0.9% NaCl solution and then stored at −80 °C. Upon use, a 1:1 (*v*/*v*) volume of distilled water was added to the RBC pellets to break the RBC membrane. RBC inclusions were separated by centrifugation at 3500× *g* for 10 min at 4 °C.

For gastrointestinal tissue samples, the stomach (ventral half of the stomach body), duodenum (first 1 cm with 1 cm distal to the pyloric sphincter), ileum (first 1 cm with 1 cm proximal to the cecum), colon (first 1 cm with 1 cm distal to the cecum) and jejunum (middle 1 cm between the duodenum and ileum) were rapidly dissected and washed with 0.1 M phosphate-buffered saline (PBS) and stored at −80 °C. For spinal cord samples, tissues were cut into three equal-length parts: spinal cord-up, spinal cord-middle, and spinal cord-down parts. Muscle samples were dissected from the femoral muscle. Other tissue samples of interest were rapidly dissected, washed with 0.1 M PBS, and stored at −80 °C.

The tissues of interest were freshly sampled for the reactive oxygen species (ROS) production assay, and freezing was avoided.

For histological analysis, mice were anesthetized and transcardially perfused with 0.1 M PBS, followed by 4% paraformaldehyde (PFA) dissolved in PBS, pH 7.4. After perfusion, tissues were collected and post-fixed in 4% PFA for 24 h and then transferred to 10% sucrose (10% *w*/*v* in 4% paraformaldehyde PBS solution) for 48 h. Afterward, the tissues were transferred to a 30% sucrose solution (30% *w*/*v* in 4% paraformaldehyde PBS solution) and allowed to sink to the bottom.

### 3.6. Immunohistochemistry (IHC) and Histopathology

The brain tissues were cut into 20-μm serial sections on a cryostat (CM3050S, Leica, Munich, Germany) and stored in PBS at 4 °C until analysis. After embedding in paraffin, the stomach (ventral half of the stomach body) was cut into 5-μm sections. Paraffin slices were subjected to a heat-mediated antigen retrieval step using citrate buffer. Then, slices were handled as previously described [29]. Sections were incubated with 1% Triton X-100 in 0.1 M PBS for 15 min at room temperature, followed by washing three times for 10 min with 0.2% Tween-20 in 0.1 M PBS (PBST). The sections were then incubated with 3% hydrogen peroxide (H_2_O_2_) for 15 min at room temperature, followed by three washes with PBST and incubation in blocking buffer (5% bovine serum albumin [BSA] in PBST) for 1 h at room temperature. The sections were then incubated overnight with selected primary antibodies (see Table 1) in PBST containing 1% BSA and 0.1% Triton X-100. After washing with PBST, the sections were incubated with the respective secondary antibodies (see Table 2) in PBST containing 1% BSA for 1 h. After incubation in 3,3′-diaminobenzidine (DAB) for 2–5 min, the slices were subjected to dehydration via an ethanol gradient. Finally, the sections were mounted with neutral balsam. Sections were viewed on an Olympus BA51 photomicroscope (Tokyo, Japan). TH-immunopositive cell numbers were counted in SN at 40× magnification using Image-Pro Plus 6.0 software (Media Cybernetics, Rockville, MD, USA).

Histopathology changes were evaluated by hematoxylin-eosin (HE) staining. Double immunofluorescence staining was performed as described above, except with different primary and respective secondary antibodies (see Table 1 and Table 2), and the sections were mounted with 90% glycerine. Sections were viewed via fluorescence laser scanning confocal microscopy (Carl Zeiss, Berlin, Germany).

### 3.7. Western Blotting

The Western blotting method was described in our previous work [29]. Tissues were weighed and homogenized in 10-fold (*w*/*v*) ice-cold radioimmunoprecipitation assay (RIPA) buffer (50 mM Tris-HCl, pH 7.5, 150 mM NaCl, 1 mM EDTA, 1% NP-40, 0.5% sodium deoxycholate, 0.1% SDS, and 0.1 mmol/L phenylmethylsulfonyl fluoride [PMSF]) with protease inhibitor cocktail (Sigma, St. Louis, MO, USA) and phosphatase inhibitors (Invitrogen, Waltham, MA, USA). The tissues were shredded into identical pieces using eye scissors, and the samples were homogenized using an ultrasonic disruption apparatus (VTX 130, Sonics, Newtown, CT, USA) on ice. The ultrasonic wave frequency was 20 kHz, and the power output was 130 W. The total ultrasonic time was 12 s, and the periods under ultrasonication and quiescence were 2 s and 10 s, respectively. The homogenized tissues were centrifuged at 12,000× *g* at 4 °C for 30 min. The protein concentrations were measured using a bicinchoninic acid (BCA) kit. Loading buffer was added to the supernatant to denature the protein, followed by 10 min of boiling, and the resulting protein lysate was stored at −40 °C. Then, 30 μg of protein from each tested group was separated on 15% gels by SDS-PAGE and transferred to a polyvinylidene difluoride (PVDF) membrane (Millipore, Burlington, MA, USA). The membrane was blocked with 3% BSA (Sigma-Aldrich, USA) and incubated with selected primary antibodies (see Table 1) overnight at 4 °C. For 5G4, anti-nitrated-α-synuclein (n-syn) and anti-phosphorylated-α-synuclein (p-syn) antibodies, and TH, after the washing step, the membrane was treated with the respective biotinylated secondary antibodies for 2 h (see Table 2) to amplify the signal. After washing, the membrane was incubated with peroxidase-labeled streptavidin (see Table 2). For other proteins, after the washing step, the membrane was treated with the relevant secondary antibodies (see Table 2). Protein bands were detected using enhanced chemiluminescence plus detection system (Molecular Device, Lmax). Densitometric analysis of the immunoreactivity of each protein was performed using image analysis software (Quantity One, Tokyo, Japan).

### 3.8. Enzyme-Linked Immunosorbent Assay (ELISA)

Tissues were homogenized and quantified as described in Section 3.7. The expression levels of mouse tumor necrosis factor-α (TNF-α) and interleukin-1β (IL-1β) were measured using commercially available ELISA kits (BD Biosciences, Pittsburgh, PA, USA) according to the manufacturer’s instructions. In detail, the first procedure was to capture antibodies. Dilute the purified anti-cytokine capture antibody to 4 µg/mL in a Binding Solution. Add 100 µL of diluted antibody to the wells of an enhanced protein-binding ELISA plate. Seal the plate to prevent evaporation and incubate overnight at 4 °C. The second step was blocking. Bring the plate to room temperature (RT), remove the capture antibody solution, and block non-specific binding by adding 200 µL of Blocking Buffer per well. Seal the plate and incubate at RT for 1–2 h. Wash more than three times with PBS/Tween. The next step was to add standards and samples (diluted in Blocking Buffer/Tween) at 100 µL per well. Seal the plate and incubate it for 2–4 hrs at RT or overnight at 4 °C. Wash more than four times with PBS/Tween. The last was antibody detection. Dilute the biotinylated anti-cytokine detection antibody to 0.5 µg/mL in Blocking Buffer/Tween. H, Add 100 µL of diluted antibody to each well. Seal the plate and incubate it for 1 h at RT. Wash more than 4 times with PBS/Tween. Dilute the streptavidin-HRP (cat. NO. 554066) to its pre-titered optimal concentration in Blocking Buffer/Tween. Add 100 µL per well. Seal the plate and incubate it at RT for 30 min. Wash more than five times with PBS/Tween. Use ABTS as a substrate. Thaw ABTS Substrate Solution within 20 min of use. Add 100 µL of 3% H_2_O_2_ per 11 mL substrate and vortex. Immediately dispense 100 µL into each well. Incubate at RT 30 min for color development. Read the optical density (OD) for each well with a microplate reader set to 405 nm.

### 3.9. Monoamine Oxidase Type-B (MAO-B) Activity Measurement

The tissues of interest were homogenized and quantified as described in Section 3.7 except that the homogenization buffer was replaced with 0.1 M PBS. As previously described, MAO-B activity was measured using a commercially available Amplex^®^ Red Monoamine Oxidase Assay Kit (Invitrogen, Thermos Fisher Scientific, Waltham, MA, USA).

### 3.10. Reactive Oxygen Species (ROS) Detection

The fresh tissues of interest were homogenized and quantified as described in Section 3.7, except that the homogenization buffer was replaced with 0.1 M PBS. ROS production was measured using a commercially available ROS detection kit (Nanjing Jian Cheng Bioengineering Institute, Nanjing, China) according to the protocol provided by the manufacturer. In detail, 10 µM 2,7-dichlorofluorescein (DCFH-DA) was added, and samples were incubated at 37 °C. Media was discarded, and samples were washed three times with a wash solution. The cell suspension was prepared by harvesting cells with trypsin. The cell suspension was centrifuged at 1000× *g* for 10 min, after which cells were collected and washed two times with PBS. ROS activity was measured at 500 nm (excitation wavelength) and 530 nm. One sample was taken from every four dams, and measurements were performed in triplicate.

### 3.11. Superoxide Dismutase (SOD) Enzyme Activity Measurement

The tissues of interest were homogenized and quantified as described in Section 3.7, except that the homogenization buffer was replaced with 0.1 M PBS. SOD activity was measured using a commercially available ROS detection kit (Nanjing Jian Cheng Bioengineering Institute, Nanjing, China) according to the protocol provided by the manufacturer. In detail, 20 µL of the serum sample and 20 µL of the enzyme working solution were pipetted to the sample well, followed by adding 200 µL of the WST working solution, then incubated at 37 °C for 20 min. Meanwhile, blank 1 (coloring without inhibitor) and blank 2 (sample blank) were prepared as indicated in the manufacturer’s manual. The absorbance at 450 nm of each well was read within 10 min on a microplate reader, and the activity of SOD was calculated with the following equation: SOD activity (U mL − 1) = (A blank1 − A sample)/(A blank1 − A blank2) × 40.

### 3.12. Measurement of Dopamine and Its Metabolites

Striatal and gastric dopamine and its metabolites were determined by high-performance liquid chromatography (HPLC) as previously described [36,37]. Briefly, tissues were homogenized in ice-cold 0.6 M HClO_4_ solution with 250 ng/mL IP as the internal standard described in Section 3.7. The homogenized tissues were centrifuged at 20,000× *g* at 4 °C for 30 min. The supernatant (120 μL) was mixed with another solution (60 μL, including 20 mM potassium citrate, 300 mM K_2_HPO_4,_ and 2 mM EDTA-2Na). After 30 min on ice, the mixed solution was centrifuged at 20,000× *g* at 4 °C for 30 min. Then, the supernatant was obtained and filtered through a 0.22-μm polypropylene nylon membrane. The filtered supernatant was injected into an HPLC system (Waters e2695, Waters, Milford, MA, USA) equipped with an electrochemical detector (Waters 2365, USA) and C18 column (4.6 mm, 150 mm; Atlantis T3; Waters, USA). The HPLC mobile phase consisted of 85 mM citric acid, 0.2 mM EDTA, 100 mM anhydrous sodium acetate, 0.5 mM octone-1-sulfonic acid and 15% (*v*/*v*) methanol in distilled water, pH 3.68. The flow rate was 1.0 mL/min. The electrochemical detection potential was 760 mV versus the Ag/AgCl reference electrode. Sensitivity was set at 50 nA full scale.

### 3.13. Statistical Analysis

The results are expressed as the means ± SEM (standard error of the mean) in each group. Statistical analysis was performed with the unpaired *t*-test using the Statistical Package for the Social Sciences (SPSS) software package (version 17.0, SPSS Inc. Chicago, IL, USA), and *p* < 0.05 was considered significant. “*n*” refers to the number of animals or tissues used from different animals.

## 4. Discussion

Protein aggregation is critical in developing neurodegenerative diseases such as Alzheimer’s disease and PD [38]. Inhibiting α-synuclein aggregation could stop disease progression in a parkinsonism mouse model, even with treatment started after disease onset [39,40]. α-synuclein is a small, soluble protein with aggregation-prone properties. Apart from the concentration of α-synuclein, PTMs, such as phosphorylation, nitration, and oxidation, can substantially impact the aggregation process [10,17,41]. Among these modifications, α-synuclein nitration is thought to play a critical role in PD onset and progression [42]. α-synuclein is a 140-amino-acid protein with four tyrosine residues, all susceptible to nitrating agents under oxidative stress conditions [30]. Once the tyrosine is nitrated, monomeric α-synuclein more easily forms dimers and further contributes to aggregation [16,30]. Therefore, protein aggregation and cellular oxidative stress are functionally connected. Evidence has suggested that high oxidative stress levels in dopaminergic neurons might be the primary cause of PD development [43].

MPTP is a lipophilic compound that could quickly pass the blood-brain barrier (BBB). In brain glial cells, MPTP is catalyzed and transformed into its toxic form, MPP^+^, which could inhibit mitochondrial complex I in neurons and glial cells, thus producing oxidative stress and leading to neuronal loss and neuroinflammation [44,45]. MPTP is conventionally viewed as a neurotoxin targeting dopaminergic neurons in the SN [46]. The peripheral effects of this compound are sometimes intentionally ignored [47]. An investigation using autoradiography to look into the distribution of MPTP revealed that the intestine has considerably high levels of radioactivity (the intensity of radioactivity indicates the amounts of MPTP or its metabolites) in the frog 24 h after MPTP injection [48]. Similar results were also observed in the C57BL/6 mouse 8 h after MPTP injection [49]. Our work demonstrated that the gastrointestinal tract naturally possesses high MAO-B activity compared to selected CNS tissues, peripheral organs, and blood components. As the crucial enzyme to catalyze MPTP into its toxic form, the relatively high activity of MAO-B in the gastrointestinal tract likely explained why the intestine has high radioactivity concentrations in these two studies. However, the localization and abundance of MAO-B vary depending on the species, as different results have been observed in mice, rats, and humans [50,51]. In the present study, besides the high activities of MAO-B, we also observed naturally low activities of SOD among selected CNS regions, peripheral organs, and blood components. We hypothesize that these two properties make the stomach vulnerable to MPTP invasion. Indeed, ROS production monitored 3 h after MPTP injection showed that the stomach appeared to undergo oxidative stress before the SN and STR.

A striking characteristic of α-synuclein discovered recently is that the peripheral α-synuclein could propagate into the CNS in a prion-like fashion [52]. Braak raised the hypothesis that the synucleinopathies induced by environmental pathogens or toxins in the peripheral organs could reach the brain via a consecutive series of projection neurons. They demonstrated that one of the most important peripheral organs was the stomach [53]. Different specially labeled forms of α-synuclein injected into the stomach, intestine, and duodenum were directly shown to propagate to the brain via the vagal nerves [13,54]. Notably, intra-gastric administration of rotenone, another mitochondrial complex I inhibitor, could reproduce PD pathology in the mice [55], and this pathology progression could be delayed by vagotomy [12]. Based on these studies, MPTP, another mitochondrial complex I inhibitor, could theoretically induce local α-synuclein aggregation in the stomach by producing oxidative stress, which could influence the brain via the vagal nerves. Indeed, in our study, we discovered that only 3 h after MPTP injection, there was already a considerable increase in the expression of the 30-kDa form of α-synuclein (possibly the dimeric form) in the stomach. At 12 h after MPTP injection, significant increases were observed in the dimeric α-synuclein (38 kDa, possibly the ubiquitinated form) and nitrated 30-kDa α-synuclein (possibly the nitrated dimeric form). In the SN and the STR, different forms of α-synuclein did not show much variation at the two points. The stomach is more vulnerable to the conventionally thought SN and STR in terms of newly appearing synucleinopathies. DA metabolism is a two-step process. MAO first metabolizes DA into 3,4-dihydroxyphenylacetaldehyde (DOPAL) and H_2_O_2_. Next, DOPAL is converted to 3,4-dihydroxyphenylacetic acid (DOPAC) by an aldehyde dehydrogenase [56]. DOPAL could bind to the Lys residues of α-synuclein [57] and be demonstrated to directly induce α-synuclein aggregation in both in vitro and in vivo models [58,59]. Therefore, the high activity of MAO-B in the stomach could lead to the aggregation of α-synuclein through converting DA to DOPAL. This mechanism may be a possible explanation for the reason for synucleinopathies induced by MPTP easily appearing in the stomach.

The MPTP/p-induced chronic PD mouse model is a long-term series of MPTP exposures, and probenecid has been used to potentiate MPTP toxicity [60]. In our studies, 10 “hits” were sufficient to induce motor and pathological symptoms resembling PD. The serial MPTP exposures may exacerbate the α-synuclein abnormalities in the stomach, further breaking down the α-synuclein equilibrium and leading to synucleinopathy propagation. Thus, we used a single MPTP injection model to explore further the mechanisms underlying why the stomach had such obvious α-synuclein abnormalities after serial MPTP injections. We noted several differences between the long-term, serial MPTP injection-induced model and the single injection-induced PD model. One difference is the severity of the synucleinopathies (See Figure 3E, Figure 10B and Figure 11D). In the chronic MPTP/p-induced PD mouse model, the levels of α-synuclein detected by anti-α-syn (C20), n-syn, and 5G4 antibodies all significantly increased. In the acute model, only the levels of a specific type of α-synuclein, detected by the anti-α-syn (C20) antibody, and the nitrated 30-kDa form of α-synuclein, were elevated, suggesting that the 5G4 antibody indeed recognized disease-related α-synuclein and that the nitrated α-synuclein was oxidative stress-related, as oxidative stress was also observed in the stomach 3 h after MPTP injection [28,61]. The second difference is the form of α-synuclein (Figure 3D and Figure 11C). We observed differences in the levels of the 55- and 25-kDa forms of α-synuclein (especially α-synuclein detected with the anti-n-syn antibody) in the stomach between the chronic model and the acute model. In the acute model, the anti-n-syn antibody mainly recognized the 38- and 30-kDa forms of α-synuclein, indicating the transient increases in nitrated dimeric and ubiquitinated α-synuclein. This dimer formation could be explained by di-tyrosine formation induced by oxidative stress [62]. In the chronic model, the anti-n-syn antibody mainly recognized the 55- and 25-kDa forms of α-synuclein, possibly suggesting that nitrated α-synuclein could further enhance α-synuclein aggregation [63]. Interestingly, a similar difference was also observed in the stomachs of two control groups (acute and chronic control groups), suggesting that aging could change the levels of nitrated α-synuclein in the stomach. Similar observations were confirmed in the SN of aging primate brains [64]. Another possibility is that probenecid may also induce α-synuclein changes in the stomach. However, thus far, no related studies have been performed. The third difference is in the cytokine profiles (Figure 7D,E and Appendix A). In the chronic PD model, our study demonstrated that DA concentrations in the stomach were elevated after serial MPTP injections, possibly mediated by the upregulated TH expression in the stomach. This phenomenon was also observed in the 6-OHDA bilateral SN-injected rat PD model [65]. Upregulated DA, TH, and D_2_ receptor expression levels may account for gastrointestinal alterations, and dopaminergic deficiency in the SN possibly induces changes in the stomach [66]. Indeed, the DA levels in our chronic model’s STR and TH-positive neurons in the SN decreased significantly. In the single MPTP injection model, we observed reduced DA levels in the stomach only 3 h after exposure in both the STR and the stomach (Appendix A). Thus, comparing these two models, pathological alterations in the MPTP/p-induced chronic PD mouse model are possibly induced by α-synuclein abnormalities rather than by the acute toxicities of MPTP. Long-term exposure to MPTP could influence the severity of the synucleinopathies and existing forms of α-synuclein, which are closely related to cytokine release and neurotransmitter changes.

In our study, only 3 h after MPTP injection, oxidative stress-induced damage had already occurred in the EGCs of the myenteric plexus of the stomach. As the so-called brain astrocytes in the gut, EGCs have long been considered mere supportive cells in the gut, similar to the previous insufficient awareness of astrocyte functions in the brain. However, accumulating evidence has highlighted their central role in regulating gut homeostasis and even brain disorders [67,68]. EGCs could be a reservoir of the prion protein transmitted from the brain to the gut or vice versa [69]. As a prion-like protein, α-synuclein has shown that it could be transmitted from the gut to the brain [13,70]. Further investigation into whether it could also propagate from the brain to the gut is required. In our study, only 3 h after MPTP injection, the EGCs were already influenced by the MPTP-induced oxidative stress, as indicated by the excessive 4-HNE production. Additionally, by this time, nitrated α-synuclein had already accumulated in the EGCs of the myenteric plexus. Owing to the unique anatomy of EGCs, which contain extensive branching and irregular processes that mingle with neuronal cell bodies and axon bundles, and the prion behavior by which α-synuclein is capable of transmitting from cell to cell [7], the α-synuclein abnormalities in the EGCs could influence the neurons in the myenteric plexus, which would cause malfunctions in the gut. The crosstalk between the EGCs and neurons has been proposed to determine transmitter release and gut motility [71]. Recently, glial connexin-43 (Cx43) hemichannels were demonstrated to play a pivotal role in the neuron-glia talk in the gut via propagating a Ca^2+^ response in the EGCs of the colon and interfering with Cx43 hemichannels slowed colonic transit [72,73]. Interestingly, α-synuclein abnormalities and Ca^2+^ influx was a mutual promotion process. Evidence showed that α-synuclein could induce Ca^2+^ influx, which could, in turn, enhance the α-synuclein toxicity [74], possibly by promoting the α-synuclein aggregation [75,76]. Although we did not monitor Ca^2+^ in the present study, given that Ca^2+^ and oxidative stress can cooperatively promote the α-synuclein aggregation [77], the alterations of neurotransmitters, fecal outputs, and other malfunctions in the stomach that we monitored in our study could be attributed to the breakdown of the neuron-glia crosstalk in the myenteric plexus induced by α-synuclein abnormalities, possibly via influencing Ca^2+^ influx and Cx43 hemichannels. However, the exact mechanisms of these processes need to be further investigated.

Our results confirmed α-synuclein abnormalities in the myenteric plexus of the stomach in the MPTP/p-induced PD mouse model. Further exploration in the acute model demonstrated that the stomach is vulnerable to MPTP invasion, producing excessive ROS, which would instantly compromise EGCs. Meanwhile, oxidative stress could induce α-synuclein abnormalities in the EGCs of the stomach, especially the nitrated form. We believe that serial MPTP exposure would enhance the α-synuclein abnormalities in the EGCs, thus influencing the neurons in the myenteric plexus. The synucleinopathies in the myenteric plexus could then propagate into the CNS in a prion-like fashion via vagal nerves. Therefore, EGCs could be a novel main contributor to synucleinopathies.

## Figures and Tables

**Figure 1 molecules-27-07414-f001:**
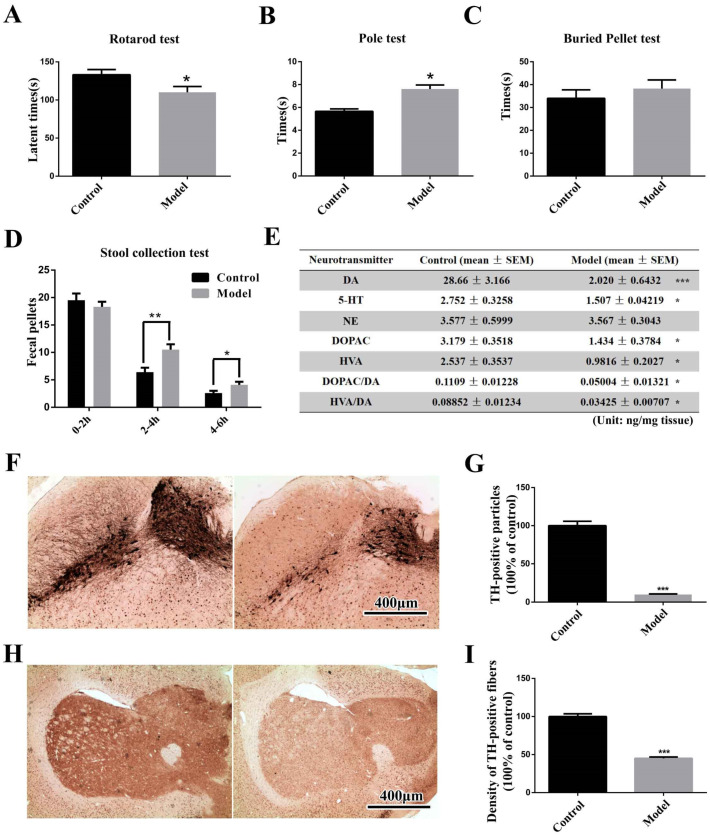
Motor, gastrointestinal and pathological alterations in chronic MPTP/p-induced PD mice. (**A**) Rotarod test performance across groups. (**B**) Pole test performance across groups. (**C**) Olfactory capacities across groups, as measured via the buried pellet test. (**D**) Stool frequency during 6 h across groups. (**E**) A table representing the expression levels of the main neurotransmitters and their metabolites in the STR across groups is shown. (**F**,**H**) Two representative photomicrographs of TH-immunoreactive particles in the SN and STR (scale bar = 400 μm). (**G**,**I**) Two histograms representing the respective quantitative analysis of TH-positive particles in the SN and STR normalized to control levels are shown. (* *p* < 0.05; ** *p* < 0.01; *** *p* < 0.001).

**Figure 2 molecules-27-07414-f002:**
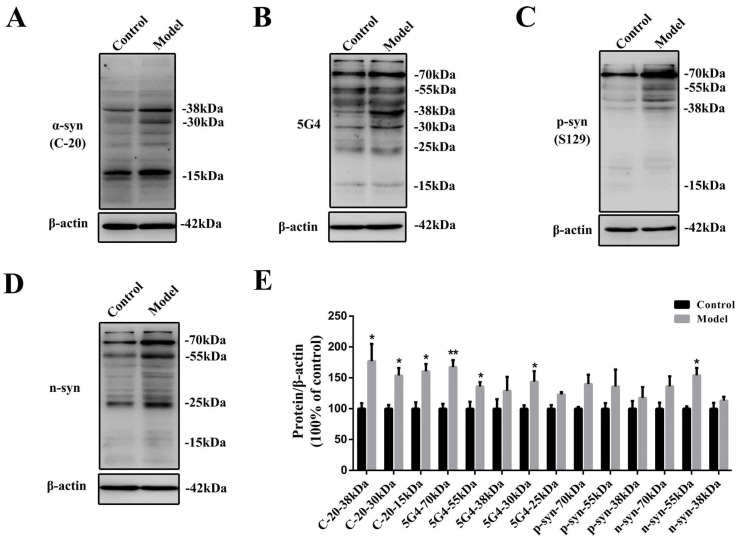
α-synuclein abnormalities in the chronic MPTP/p-induced PD mouse stomach detected by Western blotting. Representative protein bands detected using (**A**) anti-α-syn (C-20), (**B**) 5G4, (**C**) anti-p-syn (S129), (**D**) anti-n-syn and β-actin antibodies are shown. (**E**) A histogram representing the quantitative analysis of different forms of α-synuclein levels detected by anti-α-syn (C-20), 5G4, anti-p-syn (S129), and anti-n-syn antibodies normalized to β-actin protein levels is shown (*n* = 4). The data are presented as the means ± SEM, * *p* < 0.05, ** *p* < 0.01 compared with the control group.

**Figure 3 molecules-27-07414-f003:**
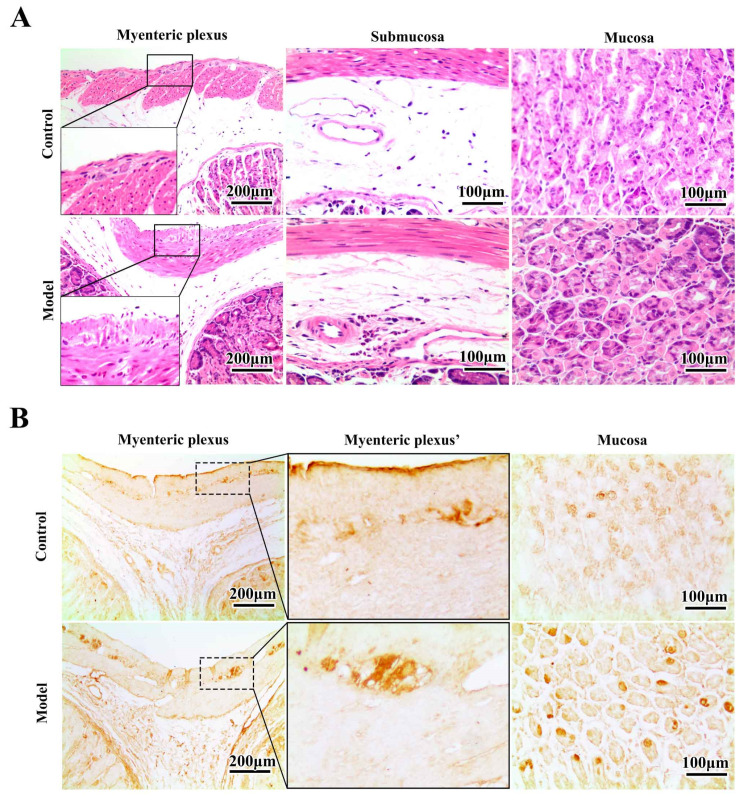
Synucleinopathy localization in the chronic MPTP/p-induced PD mouse stomach. (**A**) Representative photomicrographs of HE staining in stomachs across groups (scale bar = 200/100 μm). (**B**) Stomach sections from control and model mice were stained for oligomeric α-synuclein using the 5G4 antibody (scale bar = 200/100 μm). (**C**) Stomach sections from control and model mice were stained for nitrated α-synuclein using the anti-n-syn antibody (scale bar = 200/50 μm).

**Figure 4 molecules-27-07414-f004:**
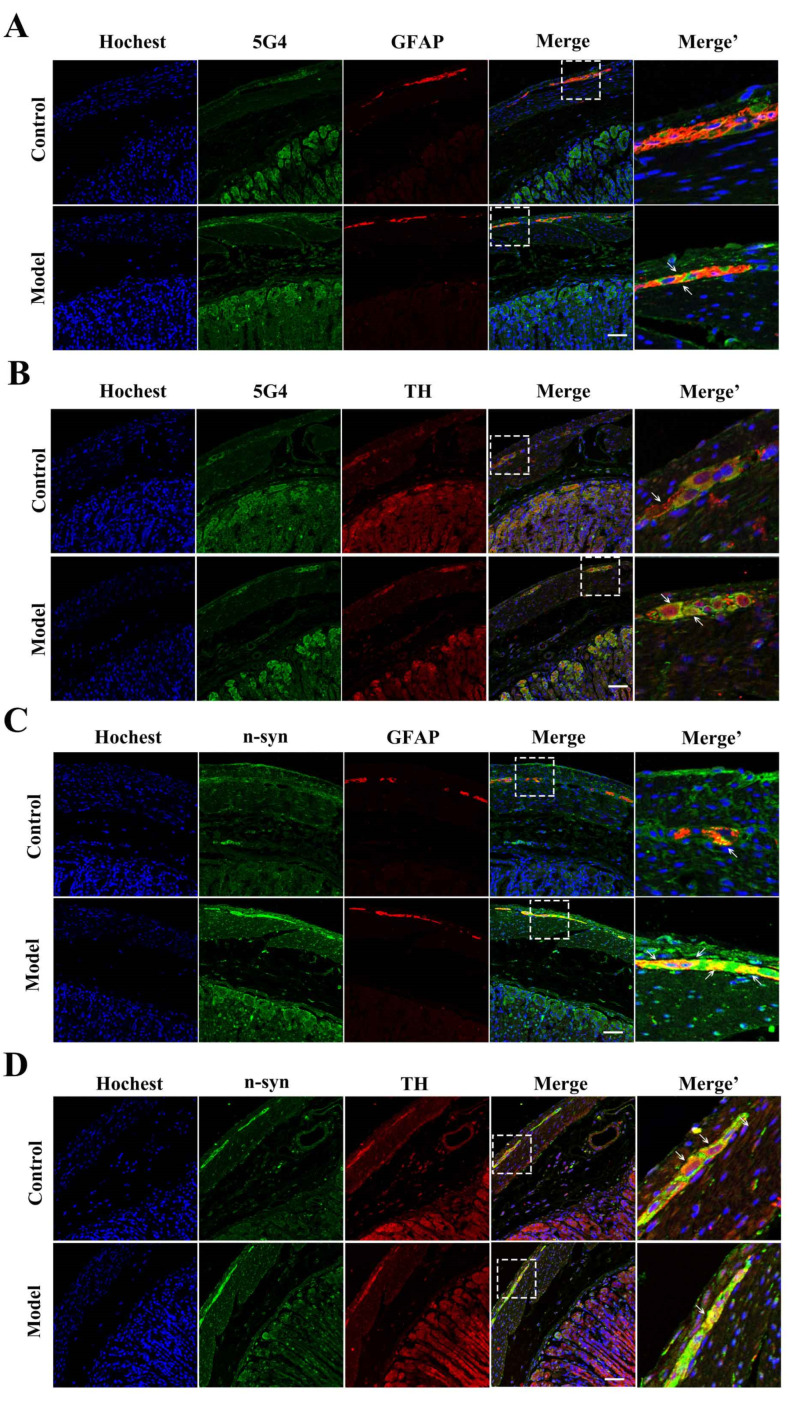
Synucleinopathies mainly existed in the EGCs and TH-positive neurons of the chronic MPTP/p-induced PD mouse gastric myenteric plexus. Oligomeric α-synuclein (detected using the 5G4 antibody) localization in stomach sections from control and model mice, additionally stained for (**A**) GFAP and (**B**) TH (scale bar = 50 μm). Nitrated α-synuclein (detected using the anti-n-syn antibody) localization in stomach sections from control and model mice, additionally stained for (**C**) GFAP and (**D**) TH (scale bar = 50 μm).

**Figure 5 molecules-27-07414-f005:**
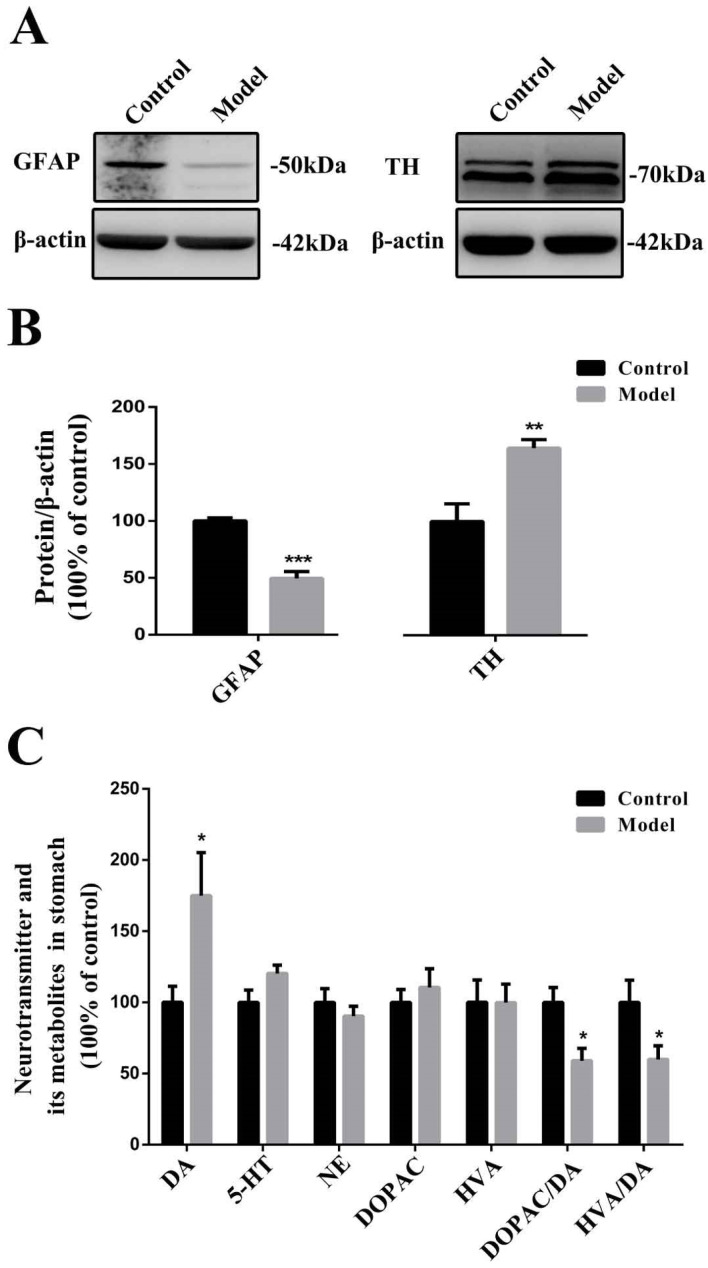
Chronic MPTP/p exposure may influence the functions of EGCs and TH-positive neurons in the stomach. (**A**) Representative protein bands detected using anti-GFAP, anti-TH, and β-actin antibodies are shown. (**B**) A histogram representing the quantitative analysis of protein levels detected using anti-GFAP and anti-TH antibodies normalized to β-actin protein levels is shown (*n* = 4). (**C**) A histogram representing the levels of the main neurotransmitters and their metabolites in the stomach across groups is shown (*n* = 7). The data are presented as the means ± SEM. * *p* < 0.05, ** *p* < 0.01 and *** *p* < 0.001 compared with the control group.

**Figure 6 molecules-27-07414-f006:**
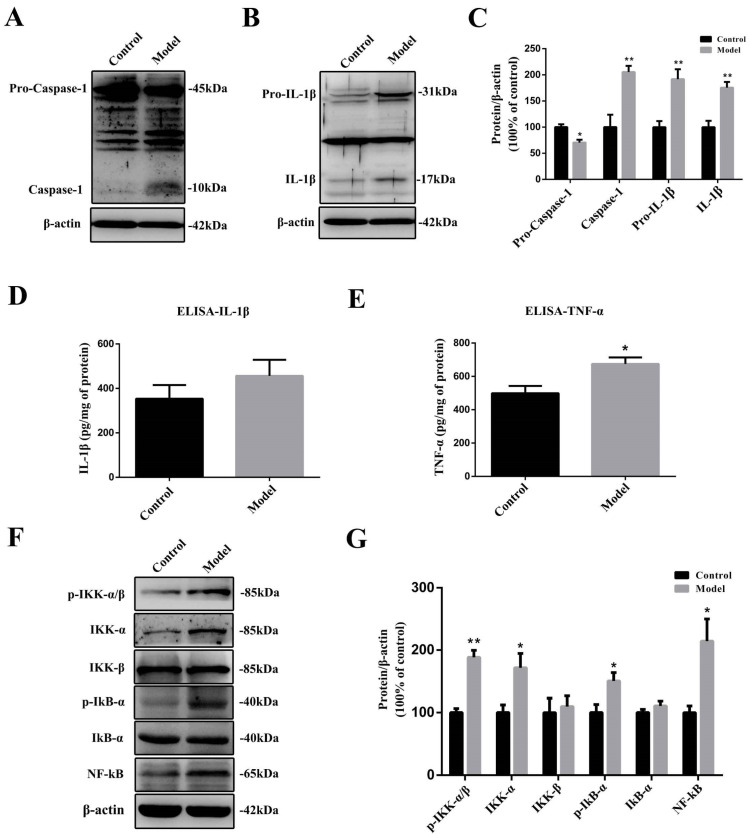
Chronic MPTP/p exposure led to a prolonged inflammatory state and activation of the NF-κB pathway in the stomach. Representative protein bands detected using (**A**) anti-caspase-1, (**B**) anti-IL-1β, and β-actin antibodies are shown. (**C**) A histogram representing the quantitative analysis of pro-caspase-1, caspase-1, pro-IL-1β, and IL-1β protein levels normalized to β-actin protein levels is shown (*n* = 4). (**D**) A histogram representing the levels of IL-1β normalized to total protein levels in the stomach measured by ELISA is shown (*n* = 4). (**E**) A histogram representing the levels of TNF-α normalized to total protein levels in the stomach measured by ELISA is shown (*n* = 6). (**F**) Representative NF-κB pathway-related molecules and β-actin protein bands are shown. (**G**) A histogram representing the quantitative analysis of p-IKK-α/β, IKK-α, IKK-β, p-IκB-α, IκB-α and NF-κB protein levels normalized to β-actin protein levels is shown (*n* = 4). The data are presented as the means ± SEM, * *p* < 0.05, ** *p* < 0.01 compared with the control group.

**Figure 7 molecules-27-07414-f007:**
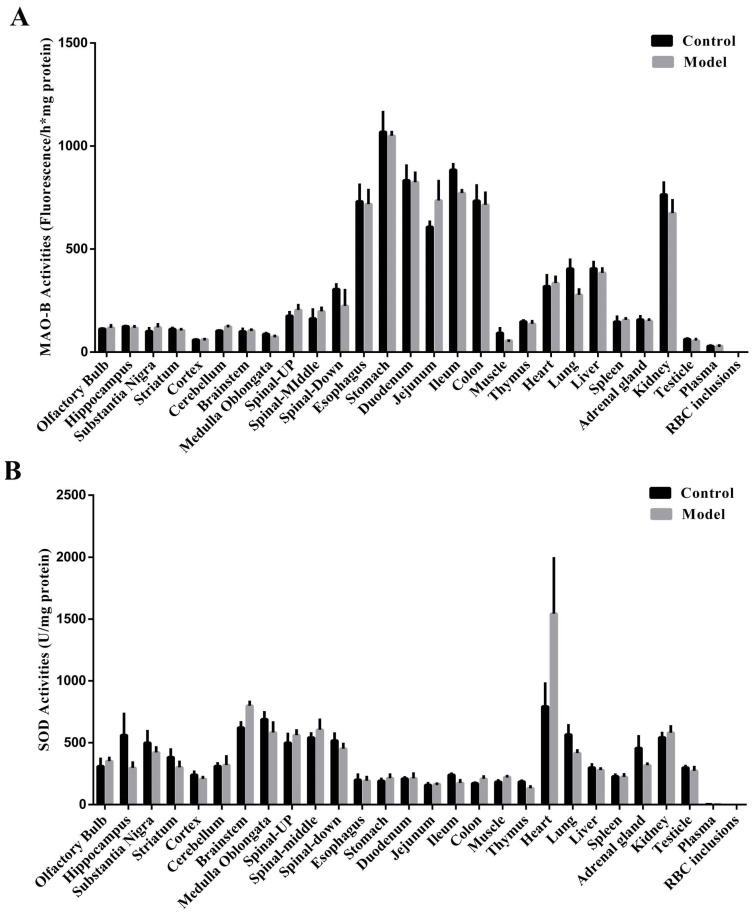
Naturally high activities of MAO-B and low activities of SOD in the stomach. (**A**) MAO-B and (**B**) SOD activities across different CNS tissues, gastrointestinal organs, blood components, and other critical peripheral organs are shown (*n* = 3). The data are presented as the means ± SEM.

**Figure 8 molecules-27-07414-f008:**
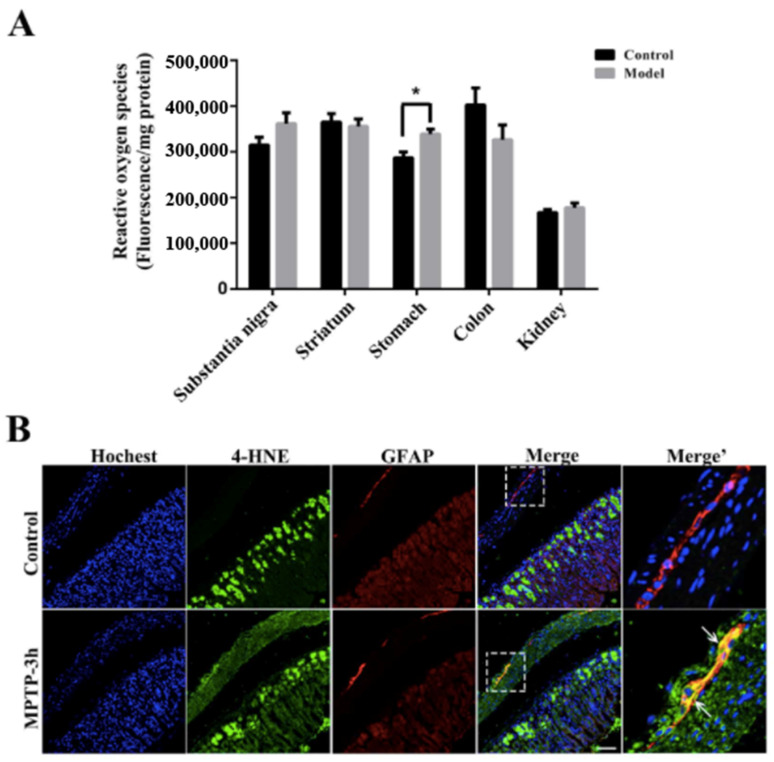
The stomach appears to undergo oxidative stress induced by MPTP before the SN, which is commonly thought to be easily affected; EGCs could be the first cell type affected by the MPTP-induced oxidative stress in the stomach. (**A**) A histogram representing ROS production across SN, STR, stomach, colon, and kidney 3 h after MPTP injection is shown (*n* = 5, * *p* < 0.05). (**B**) 4-HNE (detected by anti-4-HNE antibody) localization in stomach sections from control and model mice, additionally stained for GFAP (scale bar = 50 μm). White arrows refer to 4-HNE and GFAP co-expression in the EGCs.

**Figure 9 molecules-27-07414-f009:**
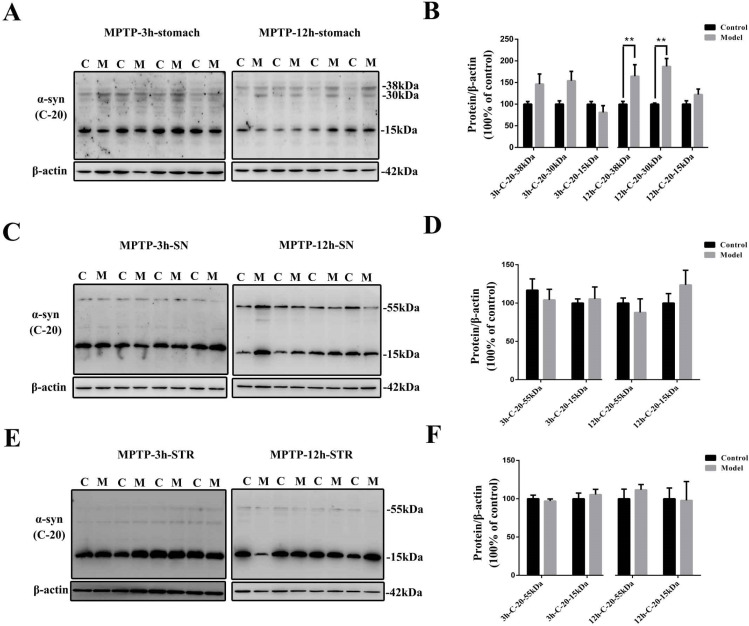
The stomach appears to develop synucleinopathies induced by MPTP before the SN and the STR. Representative protein bands of (**A**) stomach, (**C**) SN, and (**E**) STR dissected from 3 h, and 12 h MPTP-exposed mice detected using anti-α-syn (C-20) and β-actin antibodies are shown. Histograms representing the quantitative analysis of different forms of α-synuclein levels in the (**B**) stomach, (**D**) SN, and (**F**) STR from 3 h and 12 h MPTP-exposed mice detected by anti-α-syn (C-20) antibody normalized to β-actin protein levels are shown (*n* = 4). The data are presented as the means ± SEM, ** *p* < 0.01 compared with the control group.

**Figure 10 molecules-27-07414-f010:**
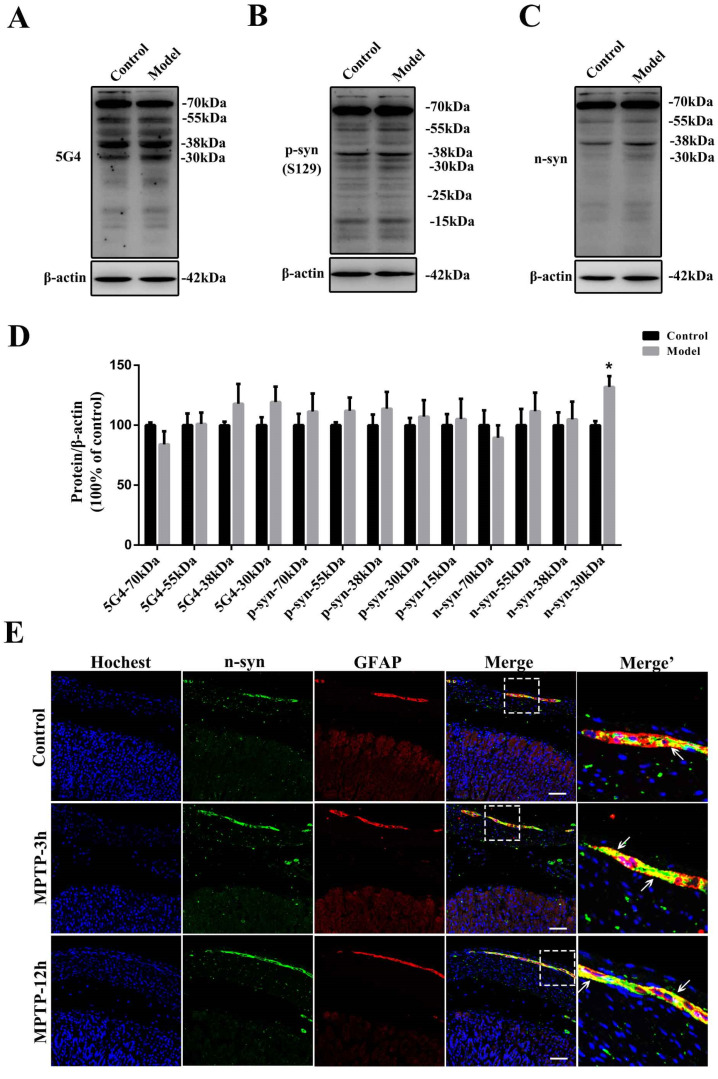
EGCs could be the initial cells contributing to the synucleinopathies. Representative protein bands of stomachs dissected from 12 h MPTP-exposed mice detected using (**A**) 5G4, (**B**) anti-p-syn, (**C**) anti-n-syn, and β-actin antibodies are shown. (**D**) A histogram representing the quantitative analysis of different forms of α-synuclein levels of stomachs dissected from 12 h MPTP-exposed mice detected using 5G4, anti- p-syn (S129), and n-syn antibodies normalized to β-actin protein levels is shown (*n* = 4). (**E**) Nitrated α-synuclein (detected using the anti-n-syn antibody) localization in stomach sections from control and model mice (3 h and 12 h MPTP-exposed mice), additionally stained for GFAP (scale bar = 50 μm). White arrows refer to n-syn and GFAP co-expression in the EGCs. The data are presented as the means ± SEM, * *p* < 0.05 compared with the control group.

**Figure 11 molecules-27-07414-f011:**
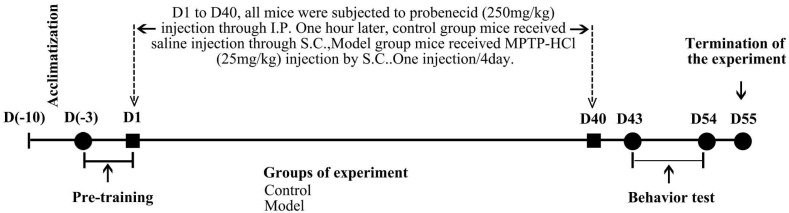
Schematic of the experimental procedure.

**Table 1 molecules-27-07414-t001:** The primary antibiodies.

Antigen	Antibody	Dilution		Source
		Western Blot	IHC/IF	
TH	Rabbit polyclonal	1:500	1:100	Santa Cruz
α-synuclein (C-20)	Rabbit polyclonal	1:500	N/A	Santa Cruz
5G4 antibody	Mouse monoclonal	1:1000	1:250	Millipore
phosphorylated-α-synuclein (Ser129)	Rabbit polyclonal	1:1000	N/A	Abcam
Nitrated-α-synuclein (nTyr125+Tyr133)	Mouse monoclonal	1:1000	1:200	Thermo Fisher
β-actin	Mouse monoclonal	1:5000	N/A	Sigma
GFAP	Rabbit monoclonal	1:1000	1:200	Abcam
Caspase-1(14F468)	Mouse monoclonal	1:500	N/A	Santa Cruz
IL-1β (H-153)	Rabbit polyclonal	1:500	N/A	Santa Cruz
p-IKK-α/β (Ser176/180)	Rabbit monoclonal	1:1000	N/A	Cell signaling
IKK-α	Rabbit polyclonal	1:1000	N/A	Cell signaling
IKK-β (2C8)	Rabbit monoclonal	1:1000	N/A	Cell signaling
p-IkB-α (Ser32)	Rabbit monoclonal	1:1000	N/A	Cell signaling
IkB-α (L35A5)	Mouse monoclonal	1:1000	N/A	Cell signaling
NF-kB (F-6)	Mouse monoclonal	1:500	N/A	Santa Cruz

N/A: Not applicable.

**Table 2 molecules-27-07414-t002:** The second antibiodies.

Antigen	Conjugation	Dilution		Source
		Western Blot	IHC/IF	
Goat anti-rabbit IgG	HRP	1:5000	1:350	KPL
Mouse IgG	HRP	1:5000	1:350	KPL
Biotinylated Anti-Mouse IgG (H+L) Antibody	Biotin	1:5000	N/A	KPL
Biotinylated Anti-Rabbit IgG (H+L) Antibody	Biotin	1:2000	N/A	KPL
Peroxidase-labeled Streptavidin	HRP	1:5000	N/A	KPL
Donkey anti-mouse IgG	Alexa Fluor 488	N/A	1:400	Life technologies
Donkey anti-rabbit IgG	Alexa Fluor 546	N/A	1:400	Life technologies
Donkey anti-goat IgG	Alexa Fluor 488	N/A	1:400	Life technologies

N/A: Not applicable.

## Data Availability

Not applicable.

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
