# Peer review of "Gastric Enteric Glial Cells: A New Contributor to the Synucleinopathies in the MPTP-Induced Parkinsonism Mouse"

_molecules, 2022, doi:10.3390/molecules27217414_

Round 1
Reviewer 1 Report
The authors of the manuscript entitled “ Gastric enteric glial cells: A new contributor to the synucle- 2 inopathies in the MPTP-induced parkinsonism mouse” aimed to investigate the acute and chronic toxic effect of MPTP subcutaneous injection in peripherical system (gastrointestinal, specially stomach) and central system (SN). The authors observed that synucleinopathy occurs firstly in stomach and followed in SN. The manuscript is well writing and it is well perfomed. It contributes to understand the early neurochemical events in peripheric which leads the neurodegerative disease's Parkinson. Follow my considerations and concerns:
Minor review
insert project number of the Ethical Commitee acceptance ;
Review the erro bars at e figures 8a and 8b;
at line 556 replace reside for residues;
during the text, italic style for in vitro and in vivo.
Author Response
Response:
We appreciate your recognition very much. We have carefully read your comments and suggestions, and we have also thoroughly addressed the points one by one as follows:
1) We have inserted project number of the Ethical Commitee acceptance (NO. 00008901);
2) We havechecked and confirmed the err bars at the figures 8a and 8b;
3) We havereplaced reside for residues at line 556in the revised manuscript;
4) We have revised italic style for in vitro and in vivo during the text in the revised manuscript.
Reviewer 2 Report
This paper describes about the mechanism underlying this phenomenon using MPTPTo find out that the enteric glial cells (EGCs) in the stomach experienced changes in terms ofα-synuclein and to to identify the mechanisms underlying this phenomenon using the single 1-
methyl-4-phenyl-1,2,3,6-tetrahydropyridine (MPTP).
the paper is well written. And the text clear and easy to read. The conclusions were consistent with the evidence and arguments presented.
The authors have studied about the Gastric enteric glial cells. However the authors need to address the following points before it can be accepted
1. Behaviour testing : Add the reference . Was nay standard protocol followed?
2. Explain ELISA in detail
3. ROS and SOD : Explain the protocol in detail.
4. Fig 5 needs to be replaced with clear images
Author Response
Response:
Thank you very much for your recognition.We have carefully read your comments and suggestions, and we have also thoroughly addressed the points one by one as follows:
1) We haveadded the reference of behaviour testing. The standard protocol was supplemented in the revised manuscript.
2) We have explained ELISA in detail in the revised manuscript.
3) We have explained ROS and SOD protocol in detail in the revised manuscript.
4) Given to our limited microscopic conditions, we did our best to provide the clear images with better resolution.